# Prompt-Agnostic Erasure for Diffusion Models Using Task Vectors

## Abstract

With the rapid growth of text-to-image models, a variety of techniques have been suggested to prevent style mimicry. Yet, these methods often only protect against specific user prompts and have been shown to allow undesirable generations with other inputs. Here we focus on *unconditionally* erasing a concept from a text-to-image model rather than conditioning the erasure on the user's prompt. We first show that compared to input-dependent erasure methods, concept erasure that uses Task Vectors (TV) is more robust to unexpected user inputs, not seen during training. However, TV-based erasure can also affect the core performance of the edited model, particularly when the required edit strength is unknown. To this end, we propose a method called *Diverse Inversion*, which we use to estimate the required strength of the TV edit. Diverse Inversion finds within the model input space a large set of word embeddings, each of which induces the generation of the target concept. We find that encouraging diversity in the set makes our estimation more robust to unexpected prompts. Finally, we show that Diverse Inversion enables us to apply a TV edit only to a subset of the model weights, enhancing the erasure capabilities while better maintaining model utility.

## 1 Introduction

The capacity of text-to-image (T2I) generative models to produce high-quality images has greatly improved over recent years. With this progress, concerns have risen over their ability to generate content that mimics copyrighted images or specific artistic styles, often without authorization. This capacity raises significant legal and ethical questions, particularly around issues of intellectual property infringement. For instance, T2I models can inadvertently reproduce images or stylistic elements that closely resemble copyrighted artworks or photographs, leading to unauthorized use. A straightforward approach to address this issue would be data filtering, removing copyrighted images and styles from the model's training dataset. However, the immense size and diversity of automatically web-scraped datasets make comprehensive filtering challenging. Imperfect filtering can compromise the legal compliance of parties deploying generative models. Moreover, even if filtering were feasible, retraining these models from scratch each time regulations change is often impractical due to the enormous costs involved.

In response, several recent methods have been proposed to sanitize unwanted concepts from T2I models, including a large set of approaches aimed at removing style mimicry and copyrighted elements (Kumari et al., 2023; Gandikota et al., 2023b;a; AUTOMATIC1111, 2022; Schramowski et al., 2023; Heng & Soh, 2023; Zhang et al., 2023a). However, these methods show significant weaknesses, particularly when subjected to unexpected inputs. Adversaries can easily bypass these methods, as shown in recent studies (Tsai et al., 2024; Pham et al., 2024). Most model sanitization techniques excel when conditioned on a specific input prompt or sequence of tokens, but they struggle with the many-to-many mapping from prompts to image space that T2I models inherently learn. Consequently, a determined adversary, even with modest computational resources, can discover alternative prompts that generate the very copyrighted content or styles that were meant to be blocked. This paper investigates the problem of eliminating unwanted concepts, specifically copyrighted images and styles, from T2I models.

While perfect erasure is beyond the scope of this study, we aim to tackle a more focused challenge. Our objective is to develop an unconditional version of concept erasure—a method that can

effectively remove copyrighted material in a manner that is agnostic to specific user prompts. Our proposed solution builds on a technique known as Task Vectors (TV) (Ilharco et al., 2023). At a high level, a task vector represents a displacement in the model's weight space resulting from fine-tuning. Previous research has shown that TVs can be used to edit a model via arithmetic operations on the model weights. Crucially, TV-based editing is independent of specific user inputs, which allows us to provide unconditional safety against the generation of copyrighted content or stylistic mimicry.

In Sec. 3, we define a criterion called unconditional safety, which measures the effectiveness of input-independent concept erasure in a model. We hypothesize that some models, given a long enough prompt, might always generate content that infringes upon copyrighted material (Wolf et al., 2023). Therefore, our proposed criterion limits the prompt to specific parameters when evaluating a method's ability to prevent unauthorized generation. Using this criterion, we demonstrate that TVs can provide unconditional safety in a controlled experimental setting.

Building on the strong performance of TV-based edits for unconditional concept erasure on our toy model, we explore whether this technique can be applied to larger T2I models without compromising their functionality on unrelated tasks. We characterize the trade-off between erasure strength and model performance using a parameter that scales the magnitude of the edit vector. To fine-tune this erasure strength, we propose a method called Diverse Inversion. Diverse Inversion identifies a broad set of token embeddings in dense space, all of which correspond to the target concept we wish to remove. This set serves as a validation set, helping us select suitable hyperparameters for our method.

**Summary of our contributions. (i)** Showing that the vulnerability of current concept erasure methods is caused by their dependence on specific input prompts (Sec. 3.2) **(ii)** Demonstrating TV-based editing as an efficient method for input-independent concept erasure (Sec. 3.3) **(iii)** Proposing Diverse Inversion, an algorithm to find a diverse set of dense prompts corresponding to a target concept, and utilizing it to allow a better trade-off between concept-erasure and model performance (Sec. 4).

## 2 RELATED WORK

**Denoising Diffusion Models.** Diffusion models are a class of generative models that iteratively refine a distribution through a Markov-based denoising process (Ho et al., 2020; Sohl-Dickstein et al., 2015). The process starts with a noise vector, $x_T$, and progressively denoises it over $T$ steps to reconstruct the original data $x_0$. Latent diffusion models (LDM) (Rombach et al., 2022) enhance the process efficiency by working in a lower-dimensional space learned by an autoencoder. The first component of LDM includes a pre-trained encoder, $\mathcal{E}$, and decoder, $\mathcal{D}$, trained on a large dataset of images. The encoder maps an image, $x$, to a spatial latent code, $z = \mathcal{E}(x)$, and the decoder reconstructs the original image from the latent code, $\mathcal{D}(\mathcal{E}(x)) \approx x$. The second component is a diffusion model trained to generate codes in the learned latent space. Given an input prompt and its associated word embedding $v$, the LDM is trained to generate an image conditioned on $c$ using the following objective function:

$$\mathcal{L} = \mathbb{E}_{z \sim \mathcal{E}(x), t, v, \epsilon \sim \mathcal{N}(0,1)} \Big[ \|\epsilon - \epsilon_\theta(z_t, v, t)\|_2^2 \Big]$$

where $z_t$ is the latent code for time $t$, and $\epsilon_\theta$ is the denoising network.

**Concept-Erasure on T2I Models.** Several strategies have been developed to prevent generative models from producing undesirable images. Negative Prompt (NP) (AUTOMATIC1111, 2022) and Safe Latent Diffusion (SLD) (Schramowski et al., 2023) suggest modifying the inference process to divert the final output from undesired concepts. Additionally, SLD consists of 4 variants SLD-Weak, SLD-Medium, SLD-Strong, and SLD-Max that correspond to erasure strength. Other approaches employ classifiers to alter the output (Rando et al., 2022; AI, 2022; Bedapudi, 2022). Since inference guiding methods can be evaded with sufficient access to model parameters (SmithMano, 2022), subsequent works including Erased Stable Diffusion (ESD) (Gandikota et al., 2023a), Selective Amnesia (SA) (Heng & Soh, 2023), Forget-Me-Not (FMN) (Zhang et al., 2023a), Ablating Concepts (AC) (Kumari et al., 2023), and Unified Concept Editing (UCE) (Gandikota et al., 2023b) advocate for fine-tuning Stable Diffusion model weights. To address additional requirements, recent works have focused on providing transferability (Lyu et al., 2024), the ability to better erase multiple concepts simultaneously (Lu et al., 2024), and improved preservation of other concepts (Zhao et al.,

2024). The idea of robust erasure was also explored by (Huang et al., 2023), which proposes a technique inspired by adversarial training.

**Jailbreaking Generative Models.** Deep neural networks are known for their brittleness and various algorithms are known for creating inputs that lead these models to produce undesirable outputs. In the context of Large Language Models (LLMs), the term "jailbreaks" refers to adversarial inputs that trigger unsafe, harmful, or unwanted responses from the model (Zou et al., 2023; Mehrotra et al., 2023; Chao et al., 2023; Wu et al., 2024). In the realm of text-to-image models, despite undergoing research on concept erasure methods used to remove undesirable concepts from the weights (Gandikota et al., 2023a;b; Kumari et al., 2023; AUTOMATIC1111, 2022; Schramowski et al., 2023; Zhang et al., 2023a; Heng & Soh, 2023), recent works have shown that they are still susceptible to adversarial inputs (Tsai et al., 2024; Pham et al., 2024). In particular, Tsai et al. (2024) uses a CLIP text encoder to construct a concept vector; a vector in embedding space representing the unwanted content. It then uses a genetic algorithm Sivanandam & Deepa (2008) to find hard prompts that produce the concept vector in the embedding space. Additionally, Pham et al. (2024) proposes Concept Inversion, which is a method based on Textual Inversion (Gal et al., 2023) to search for word embeddings that circumvent concept erasure methods. Textual Inversion (Gal et al., 2023) learns to capture the user-provided concept by representing it through new "words" in the embedding space of a frozen T2I model without changing the model.

**Task Vectors and Parameter Space Interpolations.** Although neural networks are inherently non-linear, previous research has shown that interpolating the weights of two neural networks can preserve their high accuracy if they share a portion of their optimization trajectory (Izmailov et al., 2018; Frankle et al., 2020). For example, accuracy may improve when the weights of a pre-trained model are gradually shifted towards its fine-tuned counterpart (Matena & Raffel, 2022; Ilharco et al., 2022; Wortsman et al., 2022).

Interestingly, the weight difference learned during fine tuning can also be learned on one task and transferred to another to achieve a similar function. Like a vector, it can also be multiplied by a (possibly negative) scalar, and often conveys an appropriate meaning to the model function. Ilharco et al. (2023) first compute a Task Vector (TV) as:

$$\tau = \theta_{ft} - \theta_{pre},$$

where $\theta_{pre}$ is the pre-trained model and $\theta_{ft}$ is the model fine-tuned on a selected set of tasks. Subtracting the TV, scaled by a constant $\alpha$, from the pre-trained weights $\theta_{pre}$ will make the model perform worse on the selected tasks for which the fine-tuning process was done. On the other hand, adding a scaled TV will improve the model's performance on the same tasks. Ilharco et al. (2023) show that Task Vectors can be applied to CLIP classifiers and LLMs to alter their behavior. In this work, we show that Task Vectors can also be applied to text-to-image diffusion models (in particular, the UNet module in Stable Diffusion) to perform concept erasure.

## 3 Conditional and Unconditional Concept Erasure

### 3.1 Motivating analysis

We start by noticing that current concept erasure methods are input-dependent. Such methods rely on the concept name to suppress the generation of a targeted concept. For instance, or example, ESD (Gandikota et al., 2023a) fine-tunes the pre-trained diffusion U-Net model to remove a specific concept when conditioned on a given prompt, using a loss function that reduces the likelihood of generating an image based on the concept's textual description. Since this loss depends on the concept name $c$, we hypothesize that ESD and similar methods only suppress the targeted concept when explicitly prompted with its name.

To further investigate this, we inspect the input space of the original SD 1.4 model and the same model with the Van Gogh concept removed using ESD (Gandikota et al., 2023a). In order to look for different word embeddings that would generate a target concept (e.g., "Van Gogh"), we use Concept Inversion (Pham et al., 2024) with an additional constraint used to limit it to different cosine similarity ranges from the concept name (i.e., the embedding of the string "Van Gogh"). As can be seen in Fig. 2, for the unedited model (second row, "SD 1.4") a large set of embeddings in dense space ranging in different similarities from the concept name, can all generate images featuring the

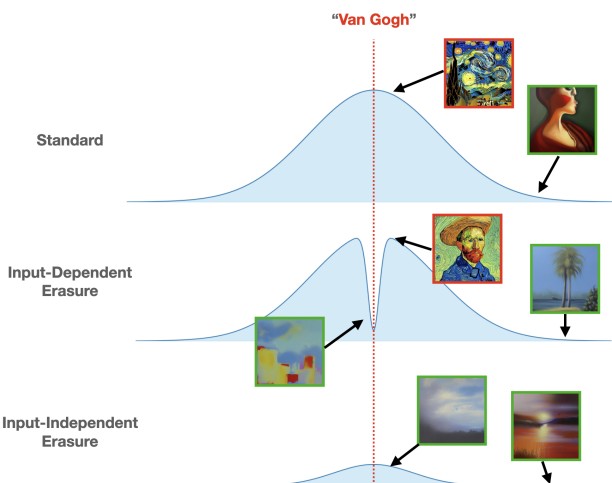

Figure 1: **Input-independent vs. Input-dependent concept erasure.** *Illustration of the probability distribution to generate the target concept "Van Gogh" across the input space. Images featuring the "Van Gogh" concept are framed in red, other images are framed in green. Input-dependent concept erasure leaves high probability areas of generating the target concept, while input-independent erasure methods erase the target concept across the entire input space.* **(Top)** *In generative T2I models, the probability of generating a specific concept is high for prompt embeddings close to the concept name, but high generation probability is possible also for prompts embedding in a significant distance from it.* **(Middle)** *Input-dependent concept-erasure attenuates the generation probability within a small environment of the given prompt but leaves a high probability of generating the erased concept further away from the prompt embedding.* **(Bottom)** *Input-independent erasure attenuates the probability of generating the target concept more consistently across the input space.*

target concept. Too far away from the concept name generations may gradually fail to reconstruct the concept. For the ESD edited model ("ESD"), the model is sanitized from the deleted concept when using the input prompt, or other embeddings very similar to it. We hypothesize that this happens because ESD only learns to filter a small neighborhood around the embedding used for training (Fig. 1). Motivated by this analysis, we turn to the main goal of this paper: First, we define a notion of safety that goes beyond a specific user prompt. Second, we suggest an effective method for concept erasure that does not depend on a specific prompt.

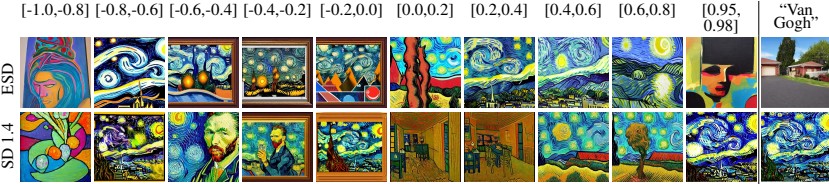

Figure 2: **Concept erasure methods often filter out only a tiny volume in input space.** *Top row: ESD (with the "Van Gogh" concept erased); bottom row: SD 1.4. We plot generations using various adversarially optimized prompt embeddings, located at different cosine similarities from the embedding of the prompt "Van Gogh". ESD continues to produce "Van Gogh" concepts when the input prompt is far away from the original concept name.*

## 3.2 MARGINAL, CONDITIONAL, AND ABSOLUTE SAFETY

A straightforward way to quantify the safety of a model against undesired generations is to estimate the marginal probability of an unwanted generation. Given a generative model $G$, a prompt $p$ drawn from a distribution $D$, and the set of unwanted generations $U$, the marginal probability $P$ for

unwanted generation is given as $S_{marginal}$:

$$S_{marginal} = P_{p \sim D}(G(p) \in U). \tag{1}$$

Yet, explicitly calculating this *marginal safety* is intractable, and depends on the unknown prompt distribution to be used during inference. It is tempting to replace it with a criterion of *conditional safety*. This notion takes into account a set of a few prompts supplied by the user, $C$, known to be related to the target concept we wish to erase. Our aim then would be to reduce the maximal probability of undesirable generation with any of the prompts $p \in C$:

$$S_{conditional} = \max_{p \in C} P(G(p) \in U). \tag{2}$$

Optimizing $G$ for this safety criteria would ensure that all prompts within the set $C$ would induce an unwanted behavior with a probability $S_{conditional}$ at most. This safety criterion is often optimized by most existing concept erasure methods. Yet, optimizing this criterion would not yield any guarantee outside the set $C$. Therefore, we focus on a safety criterion that is independent of any user-supplied prompts we term *unconditional safety*. We suggest a safety criterion limiting the probability with which undesirable generation would occur, given a constrained input complexity (e.g., the prompt length). While input length is a good parameter for input complexity, in many cases we use dense embedding. In such cases, we may instead limit the resolution in which the dense embedding is given. The resolution of a continuous input vector $v \in \mathbb{R}^d$ is closely related to the input prompt length (Wolf et al., 2023; Schwarzschild et al., 2024). For one example, encoding a higher resolution dense embedding corresponds to more bits of information (Li et al., 2008). Specifically, we denote the input complexity (measured by resolution or length) by $D_L$ and use it to write the unconditional safety criteria. Namely, the unconditional safety criterion is the maximal probability of generating the undesired content given an input of complexity $L$ (e.g., prompt length). We note this class of prompts as $p \in D_L$:

$$S_{uncoditional} = \max_{p \in D_L} P(G(p) \in U). \tag{3}$$

### 3.3 Task Vectors for unconditional safety

Although calculating the unconditional safety criterion $L_{uncod}$ is impractical for large values of $L$, we can demonstrate its improvement on toy models. We hypothesize that prompt-independent concept erasure methods such as TV edits may provide better unconditional safety. To test this hypothesis we trained a toy model with a dense "prompt" space of dimensions $d = 8$. We use images from the MNIST (LeCun et al., 2010) dataset.

We investigate the generation of MNIST digits 0 and 1 ("the target concept") from our three diffusion models: (i) *Input dependent concept-erasure*: We fine-tune the model to produce the remaining 9 digits when given 0 as conditional input. (ii) *Input-independent concept-erasure:* We utilize a TV edit for input-independent concept-erasure (Ilharco et al., 2023). We fine-tune our model to generate only the target concept (digit 0), and then subtract the Task Vector achieved by the fine-tuning process from the original model. This process is input-independent as we perform unconditional fine-tuning discarding the usage of conditional input embedding (iii) *Original model:* We also evaluate the original model, without concept erasure. For all models, we use a pre-trained classifier to automatically evaluate whether the target concept was indeed generated.

We now turn to evaluate the unconditional safety criterion for the three models above. Specifically, we define the input complexity classes $D_L$ as the resolutions in which we perform an exhaustive search of the possible prompts in continuous space. For each complexity class $D_L$, we explore a grid in $d$ dimensions, with the inspected values in each dimension comprised of $L$ equally spaced values, totaling $L^d$ points per grid. For example, the grid point $(0.0, 1.0, 1.0, 0.0, -1.0, ...)$ belongs to a low input complexity class, while the grid point $(0.4, 0.8, 0.6, -0.2, -1.0, ...)$ belongs to a higher one. Intuitively, the higher the input complexity class we examine, the closer we are to an exhaustive search in the input space.

Fig. 3 shows that the TV edit provides a much better unconditional safety $L_{uncond}$ guarantee. The original and fine-tuned models provide a non-trivial probability of generating the target concept, even for relatively low complexity parameters $L$. Moreover, for the fine-tuned concept erasure, we can find prompts in the medium complexity range ($L = 5$) that generate the target concept with very high probability. However, with the TV-based concept erasure, the undesirable generation can

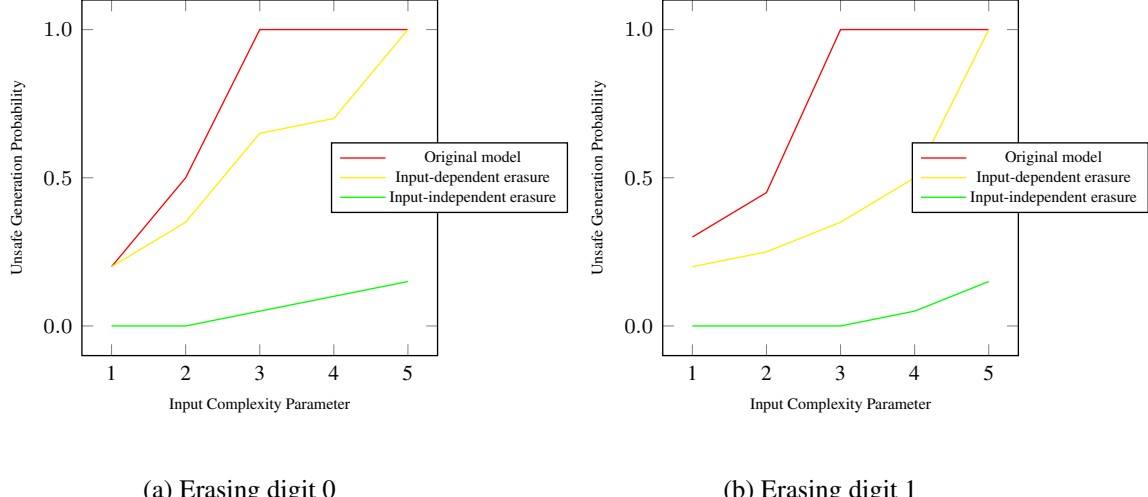

(a) Erasing digit 0               (b) Erasing digit 1

Figure 3: **TV-based concept-erasure provides better unconditional safety.** *We plot the probability of undesirable generation with the most successful adversarial prompt from each given input complexity class (See Sec.3.2). Input-independent (TV-based) erasure better reduces the probability of undesirable generations compared both to the original and the input-independent models, across the different complexity classes.*

be better mitigated across all examined complexity classes. This suggests that the TV-based erasure does not merely input-filter the model, but attenuates its ability to generate the unwanted concept more robustly across the input space. Interpreting this result according to our unconditional safety criteria (Eq. 3), we find that only the input independent method provides a non-trivial erasure of the undesirable generation probability for complexity parameters $L \geq 5$.

# 4 DIVERSE INVERSION FOR CONCEPT ERASURE USING TASK VECTORS

Motivated by the potential of TV-based editing as a method capable of improving the unconditional safety of T2I models, we now focus on applying this technique to larger models. Namely, we wish to erase unwanted concepts from large diffusion models while otherwise retaining their text-to-image capabilities. Measuring the degree of preservation of the desired text-to-image capabilities can be done directly, since this typically involves expected user inputs and outputs. However, anticipating the model's reaction to adversarial prompts *unknown* at the time of editing can be challenging. To estimate how well the model is protected against unexpected inputs, we would like to observe its generations with a diverse array of adversarial prompts. We cannot inspect all the input prompts of a given length as we did for the toy model, due to the very large number of possible prompts. To this end, we create a diverse safety validation set composed of diverse input tokens that can all generate unwanted content with the original model.

Our method for applying Task Vectors for concept erasure in large models consists of three parts. First, we learn a diverse set of adversarial embeddings, allowing us to estimate Task Vector edit robustness. Next, we show this learned set allows us to choose robust hyper-parameters for TV edits while maintaining the model utility. Finally, we show that we can not only choose performant hyper-parameter values but also sub-select the set of model parameters we wish to edit for better performance. Our motivation for utilizing Task Vectors stems from the fact that the erasure process does not depend on any prompts. The learned embeddings are only used to help us tune the editing strength of the model.

## 4.1 DIVERSE INVERSION

As we discuss in Sec. 3, concept erasure methods can provide a false sense of security by performing "input-filtering". This suggests that additional inputs are needed to better evaluate concept erasure

methods. We would like to have a diverse set on inputs, evaluating the concept erasure capability independently from any specific adversarial prompt. Therefore, these additional inputs need to be far from the embedding of the concept name as well as sufficiently diverse. $a, b$ control the desired minimal and maximal distance between the learned embeddings and the text embedding, and $c, d$ control the distance between learned embeddings pairs (see also App.B). To create such a list of inputs one can search for word embeddings as follows:

$$
\mathbf{v}_* = \arg\min_{\mathbf{v}} \mathbb{E}_{z \sim \mathcal{E}(x), v_i \sim \mathbf{v}, \epsilon \sim \mathcal{N}(0,1), t} \left[ \| \epsilon - \epsilon_\theta(z_t, v_i, t) \|_2^2 \right],
$$
$$
\text{s.t.} \begin{cases} \text{Sim}(v_i, v_{\text{concept}}) \in [a, b] \ \forall \ i = 1, 2, \ldots, n, \\ \text{Sim}(v_i, v_j) \in [c, d] \ \forall \ i, j = 1, 2, \ldots, n, i \neq j. \end{cases} \tag{4}
$$

In Eq. 4, we optimize for a set of embeddings $\mathbf{v}_* = (v_1, v_2, ..., v_n)$. The first constraint ensures that the learned embeddings are not too close to the embedding of the concept name (e.g. Van Gogh). The second constraint pushes the learned embeddings away from each other to diversify them. Nevertheless, the optimization procedure in Eq. 4 can be highly non-convex, and we found that vanilla inversion with random restarts can be used as an approximation to learn sufficiently diverse embeddings for our proposed erasure method.

### 4.2 How strong and where to edit using TV?

With the augmented set of inputs that all make Stable Diffusion generate images of the target concept, we can choose the parameter $\alpha$ that controls the edit strength of the TV. We look for a value that suppresses any such generation with any prompt from our Diverse Inversion set. Yet, a robust model might not always be usable in practice Pham et al. (2024). I.e. model outputs for non-adversarial prompts may not align with the prompt semantics. Hence, we also measure the model performance on control tasks featuring unrelated concepts. Examining both measures we can the value of the edit strength $\alpha$ and pick the $\alpha$ that yields the desired trade-off between robustness and usability (Sec. 5.2). Large values of $\alpha$ that make the Stable Diffusion model more robust against inversion might in some cases degrade the usability of the model. Motivated by Hase et al. (2023) and Maini et al. (2023), we hypothesize that we do not always need to edit all layers of the model UNet and try pruning certain blocks of the TV, i.e. setting certain values of TV to zero.

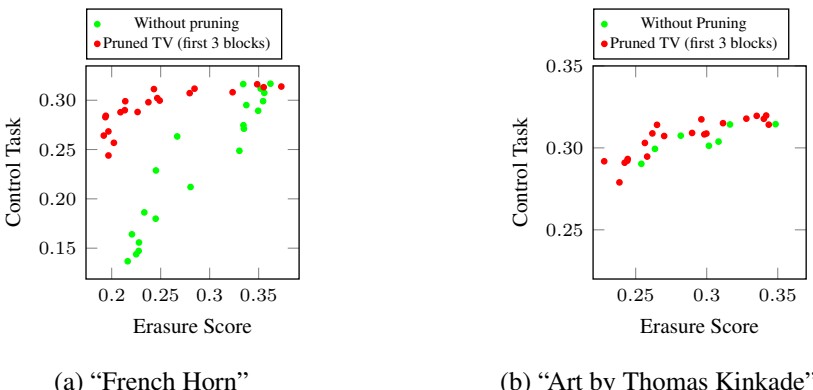

(a) "French Horn"  (b) "Art by Thomas Kinkade"

Figure 4: **The trade-off between erasure score and control task performance.** *Each scatter point will represent applying TV with different editing strength $\alpha$. We plot the robustness measured according to erasure score, **lower is better**, and control task performance, **higher is better**. Our Diverse Inversion method allows us to explore the trade-off between concept erasure robustness and model utility when editing different subsets of the model parameter.*

## 5 EXPERIMENTS

### 5.1 EXPERIMENTAL SETUP

*Metrics:* To assess the content of the generated images, we use CLIP ViT-B/32 (Radford et al., 2021) pre-trained on LAION-2B (Schuhmann et al., 2022). Following previous works Pham et al. (2024); Heng & Soh (2023); Gandikota et al. (2023a;b), the control task for all experiments is the average CLIP similarity score of 6 concepts across 3 different concept categories (artistic style, objects, and specific people): "art by Kilian Eng", "art by Picasso", "garbage truck", "chain saw", "Brad Pitt", and "Angelina Jolie".

**Erasure Score (ES).** We propose to use a metric known as *Erasure Score* to validate the robustness of the edited Stable Diffusion model to many different attack prompts. The metric is defined as follows: after obtaining word embeddings via Diverse Inversion, we generate an image for each learned embedding using the Stable Diffusion model. The Erasure Score is defined as the maximum (calculated over all generated images) CLIP similarity between the generated images from a prompt in our Diverse Inversion set, and the concept name. A lower Erasure Score indicates more robustness against adversarial inputs.

### 5.2 RESULTS

We demonstrate our method provides robustness to current adversarial methods. In Figs. 7 we demonstrate our robustness to Concept Inversion (Pham et al., 2024); first showing the original (unedited) model in the first row. In the second row in each of the sub-figures of Fig. 7 we show that for certain values of the edit strength $\alpha$, the Stable Diffusion model manages to suppress the generation of the targeted concept when explicitly prompted with the same concept name. However, when Concept Inversion (Pham et al., 2024) is applied, we can still recover the erased concept. On the other hand, when $\alpha$ is increased, we obtain both a lower Erasure Score (ES) and a more robust erased model. This suggests that we can use the Erasure Score to guide us in selecting an appropriate edit strength to make the model more robust against adversarial inputs. We also test our edited models against hard prompts obtained from Ring-A-Bell (Tsai et al., 2024), UnlearnDiffAtk (Zhang et al., 2023b), and P4D (Chin et al., 2024). Fig.6 demonstrates that our method is more robust against Ring-A-Bell compared to other concept erasure methods. For space considerations, we refer the readers to the Appendix A for the remaining results on the robustness of our method.

Additionally, to study how our method affects unrelated concepts, we calculate the MS COCO FID of the model with the Van Gogh concept erased. Tab. 1 suggests that TV-based erasure can preserve unrelated concepts competitively with other concept erasure methods. Additional results are found in App.C.3.

A notable drawback of using Task Vectors (TV) is that this method requires editing the entire model to enhance the Stable Diffusion model's robustness against adversarial inputs. Consequently, this might compromise the model's generative performance on concepts unrelated to the erased concept. Fig. 4 demonstrates that certain layers of TV can be pruned to better preserve generative performance on unrelated concepts, while maintaining robustness against adversarial inputs. Moreover, we observe that TV is worse at preserving surrounding concepts close to the concept we wish to erase. Fig. 5 shows that when erasing the Van Gogh concept, a similar concept like Monet will be more affected than less related concepts.

**Beyond style mimicry and copyrighted contents generation.** A natural question to ask is whether our method can extend to concepts other than artistic style. Using the Imagenette dataset (Howard, 2019), we tested our approach on 10 object classes, following a setup similar to Pham et al. (2024) to evaluate the robustness of erasure methods. Specifically, we generated 500 images with prompts containing the target concept and another 500 using Concept Inversion. A pre-trained classifier then measured the presence of the erased objects in these images. Table 2 shows that TV-based erasure is more robust against Concept Inversion than other methods. Our approach is particularly effective in erasing narrow concepts like artistic styles, objects, and identities. However, broader concepts, such as "Nudity" or "Party," involve a more complex set of features, making them harder to erase. While TV can erase nudity (52.12% reduction in detected body parts using the I2P benchmark), it requires a strong edit, leading to a 31.32% drop in FID score on MS COCO. Despite

this, TV-based erasure shows promise in robustly removing narrower concepts like styles and object types. We expand on this topic in App.6.

Van Gogh   Monet   Picasso

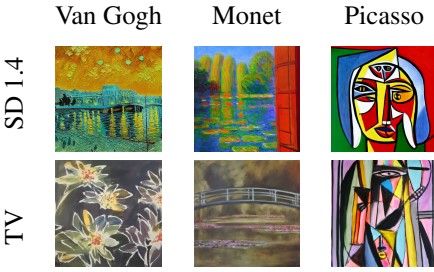

Figure 5: ***Generation of unrelated concepts***. *Images generated by the original SD 1.4 (left) and the edited model using TV (bottom). The targeted concept for TV is Van Gogh style. TV might interfere with surrounding concepts such as Monet but not further concepts like Picasso.*

Table 1: ***FID on MS COCO***: *We computed the FID score using 10k randomly sampled images from the MSCOCO validation set and generated images corresponding to the same captions using 50 steps of the DDPM sampler. SD stands for the original SD 1.4 model. TV for task vectors. AC and NP are concept erasure methods prposed by Kumari et al. (2023) and AUTOMATIC1111 (2022) respectively.*

| Method | SD | TV | AC | NP |
|--------|-------|-------|-------|-------|
| FID | 26.75 | 26.96 | 27.51 | 32.57 |

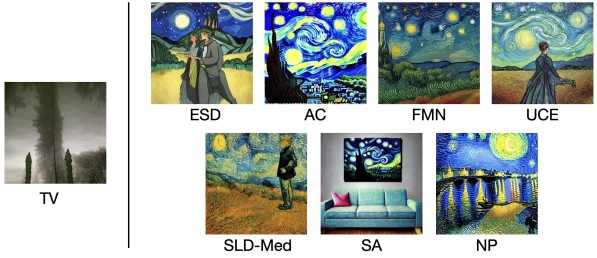

entrasplenyal courier dracing arthistory (. gogh couldn ance stars woolf
vrpaintings unbelievably about starry lives utterly ꙅ?.. vangogh motipreviously
lovers. approprireferences ♂moonlight give featuring gogh reloaded theme

Figure 6: ***Generated images with the Ring-A-Bell (Tsai et al., 2024) prompt for the concept "Van Gogh".*** *The shown adversarial prompt can circumvent 7 leading erasure methods, but not ours.*

Table 2: ***Quantitative results of Concept Inversion for different concepts (Acc. % with the given prompt / Acc. % with CI adversarial method)***: *We generate images of the erased objects, once with the given text prompt and once with an adversarial method (Pham et al., 2024). We inspect with a classifier (as in Gandikota et al. (2023a) both generations for the erased concept, and find that only the TV method provides robust concept erasure in both cases). We compare the following methods: SD (original model), TV (ours), ESD(Gandikota et al., 2023a), UCEGandikota et al. (2023b), NPAUTOMATIC1111 (2022), SLDSchramowski et al. (2023). For Safe Latent Diffusion (SLD) the original paper proposes 4 variants SLD-Weak, SLD-Medium, SLD-Strong, and SLD-Max that correspond to erasure strength. We followed previous literature and chose SLD-Med for our experiments.*

| Method | SD 1.4 | TV | ESD | UCE | NP | SLD-Med |
|--------|--------|----------|----------|-----------|-------------|-------------|
| cassette player | 6.4 | 2.0 / 0.0 | 0.2 / 6.2 | 0.0 / 2.8 | 4.0 / 9.4 | 1.0 / 2.4 |
| chain saw | 68.6 | 1.2 / 0.3 | 0.0 / 64.0 | 0.0 / 43.6 | 4.0 / 82.8 | 0.8 / 86.6 |
| church | 79.6 | 12.4 / 0.4 | 0.8 / 87.4 | 10.0 / 82.2 | 25.4 / 78.4 | 20.6 / 72.0 |
| english springer | 93.6 | 9.1 / 0.3 | 0.2 / 48.2 | 0.0 / 69.6 | 27.0 / 90.4 | 24.6 / 96.4 |
| french horn | 99.3 | 26.1 / 0.0 | 0.0 / 81.6 | 0.4 / 99.4 | 62.4 / 99.0 | 17.0 / 97.6 |
| garbage truck | 83.2 | 9.2 / 0.4 | 0.8 / 57.0 | 16.4 / 89.6 | 39.4 / 84.6 | 19.8 / 94.8 |
| gas pump | 76.6 | 3.2 / 0.3 | 0.0 / 73.8 | 0.0 / 73.0 | 18.0 / 79.6 | 12.8 / 75.6 |
| golf ball | 96.2 | 13.4 / 0.5 | 0.0 / 28.6 | 0.2 / 18.6 | 45.2 / 88.4 | 60.2 / 98.8 |
| parachute | 96.2 | 19.2 / 0.0 | 0.0 / 94.0 | 1.6 / 94.2 | 32.8 / 77.2 | 52.8 / 95.8 |
| tench | 79.6 | 12.2 / 0.1 | 0.3 / 59.7 | 0.0 / 20.6 | 27.6 / 72.6 | 20.6 / 75.4 |
| Average | 77.9 | 10.8 / 0.2 | 0.2 / 60.1 | 2.9 / 59.4 | 28.6 / 76.2 | 23.0 / 79.5 |

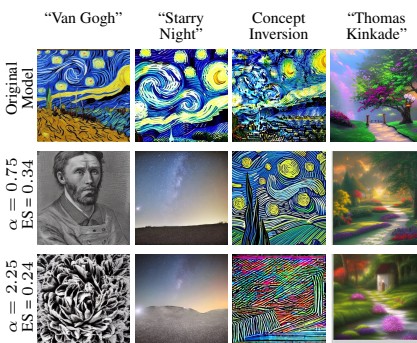 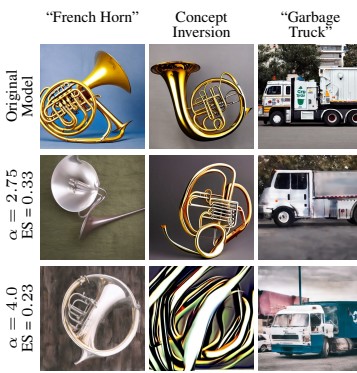

Figure 7: ***TV based concept erasure robustness to Concept Inversion.*** *A full TV edit is utilized to erase "Van Gogh"* **(Left)** *A pruned TV edit is utilized to erase "French Horn"* **(Right)**. *We display three model variants (by row): the original model, and two models from which we removed the targeted concept using Task Vectors of different magnitudes. In both cases, TV-based erasure is robust against Concept Inversion (Pham et al., 2024) and preserves the model utility on the control task (last column).*

## 6 DISCUSSION AND LIMITATIONS

**Absolute Erasure.** Making it impossible to generate unwanted content may seem like an ideal goal. However, completely avoiding unwanted content depends on the ability to recognize all unwanted behaviors. The ability to describe *all* unwanted behaviors without defining such behaviors in advance is therefore left for further research.

**Provable guarantees for erasure.** An inherent weakness of any erasure method is the inability to evaluate them in advance against yet unknown future adversarial methods (Amodei et al., 2016). We acknowledge this as a weakness of our suggested method as well. To try to mitigate this, we provide not only results against current adversarial methods but also a principled analysis of the unique qualities of TV-based concept erasure being input-independent. Yet, a full mathematical proof of the robustness is beyond the scope of this paper.

**Limitations TV-based erasure.** Our suggested method is reliant on the TV technique. Yet, the parameter space of neural networks is far from being completely understood (Ma et al., 2020). This means that the exact cases where TV-based erasure can work or fail are not completely clear. The exact limits of applicability of Task Vectors is yet to be explored.

**Selecting a concept erasure setting for various tasks.** We would like to emphasize that real-world settings differ from one another, and therefore might require different solutions. E.g., for maintaining copy-rights, robustness to unexpected prompts, which is a relative advantage of our method, might be needed. Yet, for other applications, the ability to erase broad concepts such as nudity might be more important than adversarial robustness. We believe therefore that there is room for more than one method, at least until a single method would possibly incorporate all advantages.

## 7 CONCLUSIONS

We propose adapting Task Vectors (TV), a recently proposed technique for model editing, for erasing concepts from generative models. On a range of test cases, we demonstrate how TVs can be used to sanitize undesirable concepts from text-to-image models in a way that is independent of specific user prompts. This distinguishes TV from existing methods in the literature and makes them more robust. Our method enables us to better maintain model utility while removing harmful concepts. We anticipate that our method can be extended to other models such as large LLMs and other multimodal vision-language models.

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

## A  APPENDIX

### REPRODUCIBILITY STATEMENT

We include our source code for this project in the supplementary materials to ensure that reviewers and readers can try to reproduce the experiments.

## B  ADDITIONAL IMPLEMENTATION DETAILS

For all of our experiments except the toy MNIST one, we use Stable Diffusion 1.4 (SD 1.4). To compute the TV for SD 1.4, we fine-tune the UNet component on 15 synthetic images and 15 real images obtained from Google Images (30 in total). The synthetic images are obtained from the unedited SD 1.4 using the prompt "a photo of [*object name*]" for object concepts, and "a painting in the style of [*artist name*]" for art style concepts. We fine-tune for 1000 steps using a learning rate of $1e - 05$. Running our method to erase a single concept requires about 3 hours on a single A-100 GPU. For Diverse Inversion, we optimize 20 embeddings for each interval [0, 0.2], [0.2, 0.4], [0.4, 0.6], [0.6, 0.8], [0.8, 1]. This would result in a total of 100 word embeddings. For calculating the Erasure Score, we generate one image for each word embedding that triggers the generation of the erased concept (resulting in 100 images), and 7 images for each control task prompt (resulting in 42 images).

## C  IMAGES FOR MNIST EXPERIMENT

Fig. 14 demonstrates that both Fine-tuning and TV can be used to erase the digit 0 from the toy diffusion model. We also use a pre-trained classifier to quantitatively assess the generation quality of the edited models in Tab. 4. Both editing methods can erase the target class when the model is given 0 as conditional input, while preserving generative performance on other classes. However, Fig. 8 shows that TV is more robust against inversion. For the toy diffusion models, we used implementation from Sehwag (2021). However, we modify the architecture to support conditional embeddings of dimension $d = 8$, and normalize the input embeddings during training and inference. Such modifications are made to make the space of input embeddings that generate actual digits more compact. This makes the model more likely to generate faithful images when given our sampled embeddings as conditional input. We train and fine-tune using a batch size of $512$ for $100$ epochs.

| Method | Target Class Accuracy (%) | Other Classes Accuracy (%) |
|---|---|---|
| Original | 98.2 | 98.1 |
| Fine-tuning | **1.4** | 97.3 |
| Task Vector | **0.4** | 97.3 |

Table 4: Classification accuracy (%) on generated images.

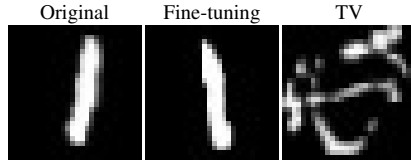

Figure 8: *Inversion of the erased class (digit 1) works on the original and fine-tuned diffusion models, but not on the edited model using Task Vector.*

### C.1 ABLATION FOR DIVERSE INVERSION

To study the necessity of Diverse Inversion, we also perform vanilla Textual Inversion (TI) Gal et al. (2023) to find 50 word embeddings for the Van Gogh style. Fig. 9 suggests that without the additional constraints, the cosine similarities between the learned embeddings through vanilla TI and the embedding of the concept name will center around 0.0. However, with Diverse Inversion, we can enhance the diversity of our learned embeddings by controlling such cosine similarities taken with respect to the concept name. Fig. 10 shows samples of SD 1.4 when we used the learned embeddings of Diverse Inversion as conditional input.

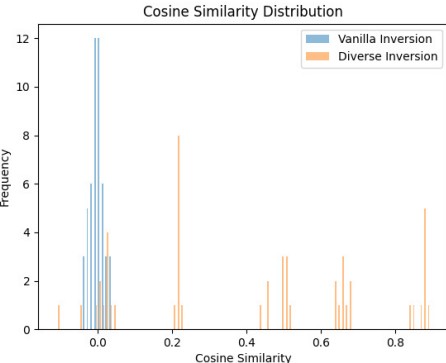

Figure 9: ***Histogram of cosine similarities between learned embeddings and the embedding of the concept name ("Van Gogh").***

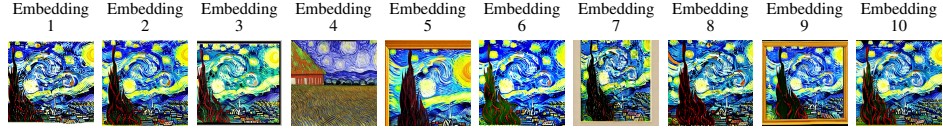

Figure 10: ***Generated images using learned embeddings from Diverse Inversion***

### C.2 ADDITIONAL BASELINE RESULTS

After obtaining the adversarial embeddings through Diverse Inversion, one baseline is simply maximizing the loss between the embeddings and its corresponding generated images. While we observe that this method can prevent the generation of Van Gogh images when prompted with the text Van Gogh, it can still be circumvented by performing another round of Concept Inversion. Using this newly learned embedding, we generate 10 images using random seeds and attain a max CLIP score of 0.33, while we get a score 0.22 on our method by applying the same procedure.

### C.3 ADDITIONAL RESULTS

**Additional attacks.** To provide additional quantitative results, we performed UnlearnDiffAtk and P4D on two erased models using ESD and TV. We computed CLIP scores across 10 runs. The results are summarized in Table 6. Qualitative samples can be found in Fig. 11.

**Multi-concept erasure.** We erase both Van Gogh and Thomas Kinkade by first taking the average of both task vectors and then we subtract it from the original model. Moreover, for each concept, we

generate 50 images and compute the average CLIP score to provide quantitative results. We report our quantitative results in Tab.6 and qualitative results in Fig.18.

**Additional qualitative results.** We provide additional qualitative results on the erasure of object identities in Fig.15. We provide results for erasure with Selective Amnesia (Heng & Soh, 2023) in Fig.17. While Selective Amnesia prevents the generation of the concept when prompted with the concept name, it is not robust against Concept Inversion (Pham et al., 2024). We provide results for Stable Diffusion 2.0 in Fig.16 that demonstrate that our method works for this model as well.

**Quantitative.** We provide additional results on pruning more blocks in Fig.21. Our claim is that the Erasure Score may allow a better exploration of the possible trade-off, rather than highlighting any specific choice.

| Concept | SD 1.4 CLIP Score | TV CLIP Score | Concept Inversion CLIP Score |
|---|---|---|---|
| Thomas Kinkade | 0.347 | 0.135 | 0.195 |
| Van Gogh | 0.282 | 0.241 | 0.233 |

Table 5: Comparison of CLIP Scores for Different Concepts.

| Method | ESD CLIP Score | TV CLIP Score |
|---|---|---|
| UnlearnDiffAtk | 0.265 | 0.211 |
| P4D | 0.258 | 0.204 |

Table 6: Concept erasure quantitative evaluation for UnlearnDiffAtk and P4D across 10 runs on ESD and TV.

Figure 11: *Generated images using ESD and TV under different attacks for style unlearning. While UnlearnDiffAtk and P4D manage to circumvent ESD, they are unable to reconstruct the erased concept from TV.*

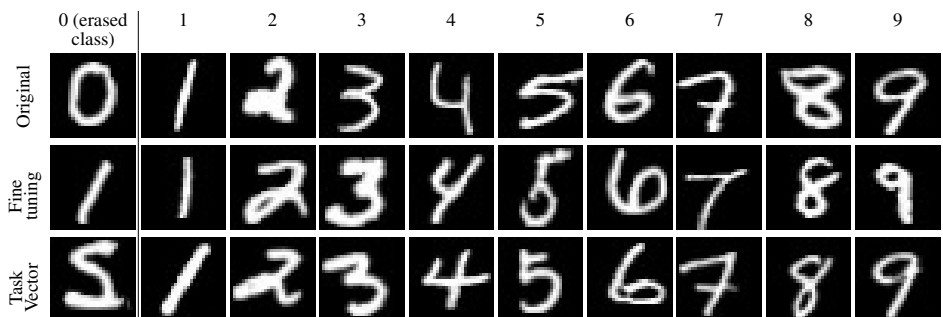

Figure 12: ***Generated images with the 'Ring-A-Bell' Tsai et al. (2024) prompt for the concept "Nudity".*** *We show that the adversarial prompt obtained from the "Ring-A-Bell" paper (bottom of the image) can circumvent 7 leading concept-erasure methods, but not our suggested TV erasure procedure.*

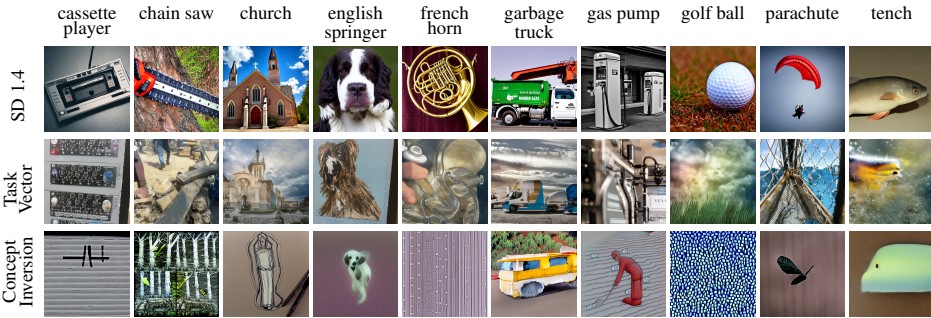

Figure 14: ***Fine-tuning and Task Vector can suppress the generation of digit 0 when the diffusion model is given class 0 as the conditional input.***

Figure 15: ***Qualitative results of Task Vector against prompts with the concepts name ("Task Vector") and prompts with Concept Inversion ("Concept Inversion") on object concepts.***

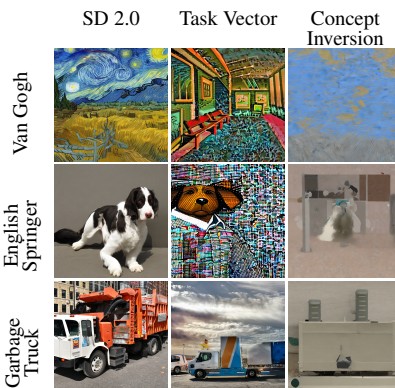

Figure 16: *Qualitative results of Task Vector concept erasure of the Van Gogh concept using Stable Diffusion 2.0; when prompted with the concepts name ("Task Vector") and with Concept Inversion ("Concept Inversion")*

Table 7: Quantitative results of Concept Inversion for different concepts (Acc. % with the given prompt / Acc. % with CI adversarial method) on Stable Diffusion 2.0

| Method | SD 2.0 | TV |
|---|---|---|
| English springer | 95.1 | 11.2 / 0.1 |
| Garbage truck | 89.2 | 8.4 / 0.2 |

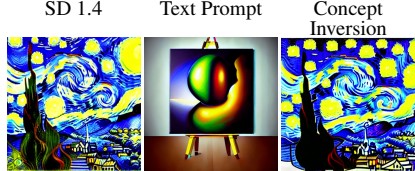

Figure 17: *Qualitative results of Selective Amnesia and Concept Inversion on Van Gogh, English springer, and garbage truck concepts using Stable Diffusion 1.40; when prompted with the concepts name ("Task Vector") and with Concept Inversion ("Concept Inversion").*

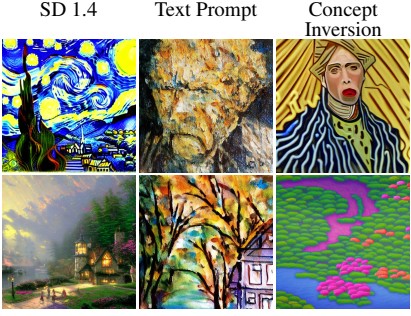

Figure 18: *Qualitative results of erasing two concepts simultaneously with the Task Vector technique - Van Gogh and Thomas Kinkade. We evaluate both erased concepts when prompted with the concepts name ("Task Vector") and with Concept Inversion ("Concept Inversion").*

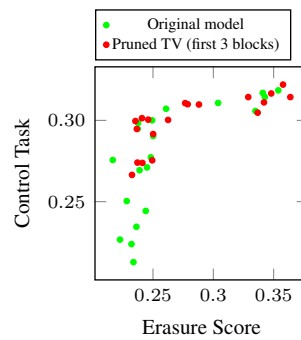
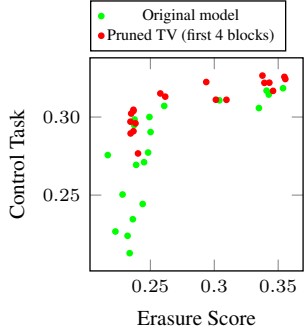

Figure 19: "Art by Van Gogh"                    Figure 20: "Art by Van Gogh"

Figure 21: *The trade-off between erasure score and control task performance. Each scatter point will represent applying TV with different editing strength $\alpha$. We plot the robustness measured according to erasure score, lower is better, and control task performance, higher is better. Our Diverse Inversion method allows us to explore the trade-off between concept erasure robustness and model utility when editing different subsets of the model parameter.*

## D  PSEUDOCODE FOR MODEL SANITIZATION

**Inputs:**

- ErasureScore Threshold - $T$
- Generative Model - $G$
- Target Concept Image Set - $S$
- Concept Description - $CONCEPT\_NAME$ and its embedding in textual space $v_{concept}$
- Learning Rate - $lr$

**Output:**

- A model sanitized from the target concept

**Steps:**

1. **Fine-Tune Model:**
Fine-tune the diffusion model's unet to generate the target concept using the set $S$ with default finetuning parameters.

2. **Calculate Task Vector:**
Compute the task vector $\tau$:
$$\tau = \theta_{ft} - \theta_{pre}$$
This represents the direction in the latent space of the unet that corresponds to generating the target concept.

3. **Find Diverse Embeddings:**
Obtain $n$ diverse embeddings $v_i$, each optimized to generate the target concept while being diverse and far from the embedding of the concept name using textual inversion (TI):
$$v_i = TI(G, S)$$

Subject to:
$$Sim(v_i^*, v_{concept}) \in [a, b] \quad \forall i = 1, 2, \ldots, n$$
$$Sim(v_i^*, v_j^*) \in [c, d] \quad \forall i, j = 1, 2, \ldots, n, i \neq j$$

For each $v_i$, optimize for generation and impose the constraints in a loop:

- Optimize $v_i$ for generation using a learning rate $lr$.

- To maintain the constraints, use a hinge loss to steer the vector back to the allowed subspace.

4. **Define Erasure Score:**
Calculate the ErasureScore as the expected value over $v_i$ ($E_{v_i}$) based on the CLIP similarity with the target concept:

$$ErasureScore = E_{v_i}(clip\_sim(G(v_i), *))$$

5. **Iterate to Find Sanitized Model:**
Loop over $\alpha$ in the range $(0, \alpha_{interval}, \alpha_{max})$:

$$\text{for } \alpha \in (0, \alpha_{interval}, \alpha_{max}):$$

- Edit the model using the task vector $\tau$:

$$G_{t_\theta} = G_{orig_\theta} - \alpha \cdot \tau$$

- Compute the similarity score:

$$SimScore = E_{v_i}(clip\_sim(G_{t_\theta}(v_i), CONCEPT\_NAME))$$

- Check if the similarity score is below the threshold. If so, return $G_{t_\theta}$.

