# OpenReview forum: "Prompt-Agnostic Erasure for Diffusion Models Using Task Vectors"
_ICLR.cc/2025/Conference — Submitted to ICLR 2025_

### Official Review · Reviewer_EQBm · 2024-10-28

**Soundness:** 2
**Presentation:** 1
**Contribution:** 2
**Rating:** 6
**Confidence:** 4

**Summary:**

This paper addresses the challenge of preventing style mimicry in text-to-image models by proposing an unconditioned approach to concept erasure, independent of user prompts. This approach uses Task Vectors (TV) for concept erasure, offering greater robustness to unexpected user inputs. Also, the authors introduce Diverse Inversion, a technique that estimates the required TV edit strength by identifying a broad set of word embeddings within the model’s input space, each capable of generating the target concept.

**Strengths:**

- Clarity and Structure: The paper is well-organized and clearly written, making it accessible and easy to follow, even for readers less familiar with the technical aspects of concept erasure and Task Vectors.
- Visualization Quality: The visualizations of generated images are well-crafted, effectively illustrating the model’s concept erasure capabilities and supporting the clarity of experimental results.
- Clear Literature Review: The related work section thoroughly covers relevant research on concept erasure and on jailbreaking generative models. This strong contextual foundation helps to situate the authors’ contributions within the broader field and underscores the necessity of robust model editing methods.

**Weaknesses:**

- Edit Block Selection: The rationale for editing the first three blocks is not fully explained. A discussion on why these specific blocks were chosen would strengthen the methodological foundation. I suggest that the authors provide a brief explanation of the model architecture and how the blocks relate to different levels of abstraction or functionality.
- Alpha Parameter Choice: The choice of α is not well-clarified. While Figure 4 mentions α, no figure or table apart from Figure 7 details the specific α values used. Since Diverse Inversion is intended to estimate the optimal strength of the Task Vector (TV) edit, it would be beneficial to provide explicit α values and clarify if the authors tested a range of α values to identify the best-performing option. I suggest that the authors include a table or figure to illustrate how they arrived at optimal strength.
- Figure Placement: Figure 1 appears on page 2, yet it is first referenced on page 4. It would improve readability and flow by moving the figure closer to its initial mention or adding an earlier reference to it in the text
- Table Clarity: In Table 2 (page 10), the acronym “SLD-Med” lacks explanation, and the term “UCE” is only briefly mentioned in the related work section (page 3). It’s unclear if SLD-Med and UCE refer to the same concept; clearer definitions would enhance comprehension. I suggest that the authors include a brief explanation of these terms in a footnote or in the table caption.
- Equation Definition: In Equation 4, the variables [a, b] and [c, d] are not clearly defined. While the meaning can be inferred from the surrounding text (Lines 341-343), each variable in the equation should be explicitly defined. I suggest that the authors consider adding a brief explanation of these variables immediately following the equation, which would maintain the mathematical formalism while improving readability. Alternatively, consider replacing the equation with a detailed textual description if it enhances clarity.
- Typos and Formatting Issues:
  - Line 285: "Sec.3.2" should be "Sec. 3.2".
  - Line 343: "e.g. Van Gogh" should be "e.g., Van Gogh".
  - Line 354: "I.e." should be formatted as "I.e.," or, for clarity, replaced with "For example,".
  - Line 355-356: The sentence lacks a verb; it currently reads “we can the value of the edit strength α.” Please revise for clarity.
  - Line 360: "i.e. setting" should be "i.e., setting".
  - Line 400: "In Figs" should be "In Fig".

**Questions:**

- Edit Block Selection: What was the rationale for choosing to edit only the first three blocks in the model? Would the authors consider expanding on why these specific blocks were selected for editing?
- Alpha Parameter Choice: The choice of the α parameter remains somewhat unclear, with few details provided outside of Figure 7. Could the authors specify the α values used throughout the experiments and clarify whether they evaluated multiple α values to determine the optimal edit strength?
- Figure Placement: Would the authors consider moving Figure 1 closer to its first reference on page 4 to improve readability and flow?
- Table Clarity: Could the authors clarify the meaning of “SLD-Med” in Table 2 (page 10) and confirm if it is the same as “UCE” mentioned briefly in the related work section? Including these definitions would improve comprehension.
- Equation Definition: In Equation 4, the terms  and  are not clearly defined. Could the authors provide explicit definitions for each variable, or alternatively, replace the equation with a detailed textual description if that would improve clarity?
- Typos and Formatting: There are minor typos and formatting inconsistencies (e.g., “Sec.3.2” instead of “Sec. 3.2”). Would the authors consider addressing these issues to enhance overall readability?

---

> ### Author Response · Authors · 2024-11-22
>
> Thank you very much for your detailed review. We appreciate that you acknowledged our motivation for robust model editing methods, and found our paper well-organized and clearly written. We address each of your questions below:
>
> ***“... a brief explanation of the model architecture and how the blocks relate to different levels of abstraction or functionality.” “What was the rationale for choosing to edit only the first three blocks in the model? Would the authors consider expanding on why these specific blocks were selected for editing?”***
>
> The UNet of Stable Diffusion 1.4 consists of an encoder with 12 blocks, a middle block, and a skip-connected decoder with 12 blocks. Motivated by [1], we hypothesize that the Task Vector might contain some noise and not all weights should be edited. Works such as [2] highlight the different functions of different layers. Namely, some Unet blocks relate to more abstract concepts, while others relate to colors, textures, or compositions. Our experiments suggest that we can set certain weights of the Task Vector to zeros to achieve better robustness versus utility trade-off.
>
> We provide additional results on pruning more blocks in Appendix C. Our claim is that the Erasure Score may allow a better exploration of the possible trade-off, rather than highlighting any specific choice.
>
> ***“Alpha Parameter Choice: The choice of α is not well-clarified. While Figure 4 mentions α, no figure or table apart from Figure 7 details the specific α values used.”***
>
> After obtaining the diverse set of embeddings through Diverse Inversion. We then look through the images that give the highest Erasure Score to and pick the alpha value that yields sufficient erasure. We observe that for most objects, an alpha between 1.25  to 1.75 is sufficient for robust erasure. On the other hand, we use an alpha between 2.0 and 2.5 for styles.
>
> ***“ It would improve readability and flow by moving the figure closer to its initial mention or adding an earlier reference to it in the text” , “Equation Definition: In Equation 4, the variables [a, b] and [c, d] are not clearly defined.”***
>
> We thank the reviewer for pointing this out, we revised the manuscript accordingly.
>
> ***“ Could the authors clarify the meaning of “SLD-Med” in Table 2 (page 10) and confirm if it is the same as “UCE” mentioned briefly in the related work section?”***
>
> SLD stands for Safe Latent Diffusion [3] which is an inference-guiding-based concept erasure method. In particular, this method modifies the inference process to divert the final output from undesired concepts. The original paper proposes 4 variants SLD-Weak, SLD-Medium, SLD-Strong, and SLD-Max that correspond to erasure strength. We followed previous literature and chose SLD-Med for our experiments. UCE stands for Unified Concept Editing [4] which is a fine-tuning-based method for concept erasure. We will clarify the meaning of SLD-Med in the main text.
>
> We thank the reviewer for their detailed suggestion and revised the manuscript accordingly.
>
> Thank you very much for your comments. We respectfully ask that if you feel more positively about our paper, please consider updating your score. If not, please let us know what can be further improved; we are happy to continue the discussion any time until the end of the discussion period. Thank you!
>
> **References:**
>
> [1] Prateek Yadav, Derek Tam, Leshem Choshen, Colin A. Raffel, Mohit Bansal. “TIES-Merging: Resolving Interference When Merging Models”. NeurIPS 2023
>
> [2] Viacheslav Surkov, Chris Wendler, Mikhail Terekhov, Justin Deschenaux, Robert West, Caglar Gulcehre. "Unpacking SDXL Turbo: Interpreting Text-to-Image Models with Sparse Autoencoders.", Preprint 2024
>
> [3]  Patrick Schramowski, Manuel Brack, Björn Deiseroth, Kristian Kersting. “Safe Latent Diffusion: Mitigating Inappropriate Degeneration in Diffusion Models”, CVPR 2023
>
> [4] Rohit Gandikota, Hadas Orgad, Yonatan Belinkov, Joanna Materzynska, David Bau. "Unified Concept Editing in Diffusion Models", WACV 2024

---

> ### Author Response · Authors · 2024-11-27
>
> Dear Reviewer,
>
> We sincerely appreciate your valuable feedback and the time you have dedicated to reviewing our submission. Your insights have been instrumental in shaping the final version of our submission.
>
> We would like to kindly remind you that the discussion period is set to conclude on December 2nd. If there are any additional questions, concerns, or clarifications, we would be delighted to continue the discussion.
>
> Thank you once again for your attention. We look forward to hearing from you!

---

### Official Review · Reviewer_yv46 · 2024-11-01

**Soundness:** 3
**Presentation:** 4
**Contribution:** 3
**Rating:** 8
**Confidence:** 4

**Summary:**

The paper presents a novel method for concept erasure in pre-trained generative models. This method consists of two key components: (1) the development of a Task Vector Method for concept erasure; and (2) the selection of optimal parameters through novel Diverse Inversion procedure. Notably, this approach is input-independent and does not rely on specific pre-defined prompts that contain concepts. As a result, it demonstrates enhanced robustness against concept inversion when compared to previous methods, while maintaining comparable results on unrelated concepts generation tasks and within the "given prompt generation" setting.

**Strengths:**

- The authors clearly identify the problem of “input dependence” associated with previous methods and provide compelling evidence of these issues via the MNIST toy experiment, which emphasizes prompt complexity rather than using a fixed set of prompts.

- They propose a method to address these challenges, which combines an existing concept-forgetting technique Task Vectors with a novel procedure called Diverse Inversion to optimize parameter selection for Task Vectors.

- Although Task Vectors is an already existing technique, the authors unveil its previously unexplored property of Concept Inversion Robustness.

- The Diverse Inversion idea is an interesting approach that could be applied to other research areas, potentially enhancing our understanding of concept learning and erasure processes.

- Overall, the text is straightforward and presents all ideas clearly and concisely.

**Weaknesses:**

- Certain aspects of the experimental workflow are not sufficiently detailed. For instance, the setup of the toy experiment on MNIST lacks information regarding the embedding grid search procedure. Additionally, the Diverse Inversion Set selection procedure may need more clarification, particularly regarding the number of restarts of the Concept Inversion procedure and a comprehensive step-by-step description.

- Furthermore, it appears that the vector from the Diverse Inversion set, which is utilized for selecting the parameter alpha, was also employed in evaluating the robustness of the methods against Concept Inversion. If this is the case, it would be helpful to report how the metrics would be affected if this vector were removed from the Diverse Inversion set.

- It would be beneficial to include additional visual examples to illustrate the results presented in Table 2.

**Questions:**

1. Was the vector from the Diverse Inversion set used in evaluating the robustness of the methods against Concept Inversion? If so, could you please provide information on how the metrics would change if this vector were excluded from the Diverse Inversion set?

2. Could you provide a step-by-step description of the Diverse Inversion Set selection procedure? Additionally, please include details on the number of restarts for the Concept Inversion procedure.

3. Why is the Control Task not utilized for selecting alpha, alongside the Diverse Inversion set?

4. Can you elaborate on the toy example, specifically regarding the embedding grid search procedure?

5. It would be beneficial to include additional visual examples to illustrate the results presented in Table 2.

---

> ### Author Response · Authors · 2024-11-22
>
> Thank you very much for your detailed review. We appreciate that you found our ideas interesting and novel, and even potentially enhancing the understanding of concept learning and erasure processes. We address each of your questions below:
>
> ***“Certain aspects of the experimental workflow are not sufficiently detailed.”***
>
> During Diverse Inversion, we set a and b in Equation 4 to be 0.1 and 0.15. Additionally, we optimize 20 embeddings for each [c, d] interval [0, 0.2], [0.2, 0.4], [0.4, 0.6], [0.6, 0.8], [0.8, 1]. These 5 runs would result in a total of 100 word embeddings. In some cases, Concept Inversion without the constraints can still lead us to pick the same alpha value as Diverse Inversion. In this case, we would run Concept Inversion 5 times, each time optimizing 20 embeddings simultaneously.
>
> ***“... it appears that the vector from the Diverse Inversion set, which is utilized for selecting the parameter alpha, was also employed in evaluating the robustness of the methods against Concept Inversion”***
>
> We would like to emphasize that when evaluating the robustness we are always using the input most successful in attacking a given defense, so there was no effect on the evaluated robustness of our method. Namely, adding an input that our method is robust against to the evaluation set does not affect its robustness scores. We validated this claim empirically.
>
> ***“It would be beneficial to include additional visual examples to illustrate the results presented in Table 2.”***
>
> We thank the reviewer for the suggestion. We provide additional results for objects in Section C.3 of the Appendix.
>
> ***“Why is the Control Task not utilized for selecting alpha, alongside the Diverse Inversion set?"***
>
> The Control Task can certainly be used to select the value of alpha. While the Erasure Score measures the concept erasure performance, the Control Task score measures the preservation of unrelated concepts. Different users may use both scores simultaneously to choose their optimal point on the offered trade-off by each method (See Fig.4 in the paper).
>
> As our focus is on concept erasure, in our case we first used the Erasure Score to choose a minimal strong enough magnitude \alpha, and then validated that this value indeed preserves the Erasure Score for the examined concepts.
>
> ***“Can you elaborate on the toy example, specifically regarding the embedding grid search procedure?”***
>
> First, we train a generative model for MNIST, conditional on a continuous input space of dimension d=8. We then perform “unconditional” fine-tuning on the digit we want to erase through further training but discard the conditional input embeddings. The fine-tuned model is combined with the original model to compute the Task Vector. The Task Vector is then used to erasure that concept.
>
> After training the models, we evaluate for each complexity parameter L the attack success rate. Namely, we check the probability of generating the supposedly erased concepts. For each value of L, we inspect all the possible conditional inputs (vectors of dimension d=8), of the form:
> $\left(\frac{2l_0 - L}{L}, \frac{2l_1 - L}{L}, \dots, \frac{2l_d - L}{L}\right)$
>
> where each input vector has d dimensions, and each $l_i \in [0 .. L]$.
>
> As we increase L we inspect more fine-grained options for the conditional input. Intuitively, as $L \to \infty$
>  we approach “exhaustive search” on the model conditional input.
>
> Thank you again for your excellent comments. We respectfully ask that if you now feel even more positively about our paper, to consider slightly increasing your score. We are happy to continue the discussion at any time until the end of the discussion period. Thank you!

---

> > ### Comment · Reviewer_yv46 · 2024-11-26
> >
> > Thank you for your explanations. I would like to clarify a few more details:
> >
> > 1. Is it true that the Concept Inversion Attack is trained on top of the sanitized model, specifically as defined by
> > $$v_{CI} = TI(G_{t_{\theta}}, S)$$
> >
> > 2. Can you describe how you obtain "the vector with the nearest vector that obeys the constraints" in the Diverse Set Construction?

---

> > > ### Author Response · Authors · 2024-11-26
> > >
> > > Thank you for acknowledging our rebuttal and continuing the discussion!
> > >
> > > Please find our answers below:
> > >
> > > 1. Yes, this is correct. After applying our method for erasure to the given model, we evaluate the model’s robustness by performing concept inversion on the sanitized model.
> > >
> > > 2. We use a hinge loss to enforce the constraints in Eq.4. Thank you for pointing out that the language in this part may be unclear. We have updated it in the revised manuscript.

---

> > > > ### Comment · Reviewer_yv46 · 2024-11-28
> > > >
> > > > I raise my grade to (8). The authors have answered all the questions and provided additional experiments. The paper presents an interesting set-up of a toy experiment on MNIST and new definitions and analyses of unconditional safety in terms of prompt length. I also like the idea of constructing a Diverse Inversion as some way to cover the entire domain responsible for a concept in the embedding space, rather than using one single vector. It could be seen as a more enhanced version of the Concept Inversion and motivate researchers to check the robustness of erasure methods not only with respect to Concept Inversion but also with respect to Diverse inversion.
> > > >
> > > > As for the object erasure method itself, it is not new, but Concept Inversion, for example, is also just Textual Inversion applied in a new problem statement. In my opinion, the article contains enough new ideas and new analyses useful for the scientific community, besides the method itself.
> > > >
> > > > However, I still recommend the authors to be more careful in describing some parts of the experiments.  It would be helpful to clarify the loss functions used, address the reviewers' questions about unconditional safety and prompt length in detail, and expand the additional experiments to include more concepts in revised version. This is essential to ensure that methods and results are fully reproducible. For the same reason, I also encourage authors to release the code in the future.

---

> ### Author Response · Authors · 2024-11-28
>
> Thank you for acknowledging our rebuttal and for taking the time to review our work!
>
> As the revision period is already over, we would like to assure you that we will incorporate the feedback from all the reviewers in the final version of the manuscript. Moreover, we will include additional concepts, as well as the code for our method.
>
> Thank you once again for a very dedicated review.

---

### Official Review · Reviewer_kjLz · 2024-11-01

**Soundness:** 2
**Presentation:** 2
**Contribution:** 2
**Rating:** 6
**Confidence:** 3

**Summary:**

The paper introduces a technique for erasing concepts from diffusion models. The method is based on using task vectors to erase the concepts, in combination with diverse inversion, a form of textual inversion. A key feature is that the erasure is prompt-agnostic and is designed to work with diverse prompts, especially adversarial ones.

**Strengths:**

* There is a decent initial analysis to motivate the approach and explain why it may be suitable.
* The method seems to perform well, maintaining the quality of the generated images for non-erased concepts, and successfully erasing the selected concepts.
* In general the paper is easy to follow.

**Weaknesses:**

* The evaluation is quite limited, it would be good if quantitative evaluation included diverse adversarial techniques in addition to concept inversion. There are some qualitative results for UnlearnDiffAtk and P4D in the appendix, but the paper would benefit from using these and maybe even others for more extensive quantitative evaluation. Also it would be good to show the method works also on other models than Stable Diffusion v1.4 specifically.
* The method seems to be primarily a combination of task vector technique and a version of text inversion, applied to the problem of concept erasure, so it may lack significant novelty.
* There are quite a few issues with the writing and presentation - the font is different than the standard one, this should be corrected; various typos, grammar issues or missing words, e.g. “jailbraking” L145, “might in some cases the usability might degrade” L358,  “Fig. 6 demonstrate” L410, “how how” L414, …

**Questions:**

* What could the prompts look like for a given complexity class L? Does it directly translate to the number of words?
* Can this method actually remove small parts of the image such as copyright logos? It was used in motivation but seems to not be tested?
* How well does the method work when using other adversarial techniques such as UnlearnDiffAtk and P4D - quantitative evaluation, not only qualitative that is already provided?
* Does the approach work well also on other diffusion models than Stable Diffusion v1.4?

---

> ### Author Response · Authors · 2024-11-22
>
> Thank you for your detailed review. We appreciate that you recognized our motivating analysis and found our paper is easy to follow. We address each of your questions below:
>
> ***“...it would be good if quantitative evaluation included diverse adversarial techniques in addition to concept inversion…”***
>
> We thank the reviewer for their suggestion. To provide additional quantitative results, we performed UnlearnDiffAtk and P4D on two erased models using ESD and TV. We computed CLIP scores across 10 runs. The results are summarized in the table below:
>
>
> | Method         | ESD CLIP Score | TV CLIP Score |
> |-----------------|----------------|---------------|
> | UnlearnDiffAtk | 0.265          | 0.211         |
> | P4D            | 0.258          | 0.204         |
>
>
> ***“it would be good to show the method works also on other models than Stable Diffusion v1.4 specifically”***
>
> We thank the reviewer for their suggestion. We provide additional results on Stable Diffusion 2.0 in Section C.3 of the Appendix.
>
> ***“The method seems to be primarily a combination of task vector technique and a version of text inversion”***
>
> We would like to highlight identifying the prompt dependence as the source of vulnerability in previous concept erasure methods. This is a main contribution of our work.
>
> ***“issues with the writing and presentation”***
>
> We thank the reviewer for pointing this out. All editorial issues are addressed in the revised version.
>
> ***“What could the prompts look like for a given complexity class L? Does it directly translate to the number of words?”***
>
> For standard Text-to-Image models the complexity class L indeed corresponds to prompts of length of L tokens.
>
> As in our toy experiments a dense input space for conditional generation (instead of text), we define the complexity class L as containing inputs of the form:
>
> $\left(\frac{2l_0 - L}{L}, \frac{2l_1 - L}{L}, \dots, \frac{2l_d - L}{L}\right)$
>
> where each input vector has d dimensions, and each $l_i \in [0 .. L]$.
>
> Namely, higher complexity classes correspond to more specification options in the input space of the conditional generation; this is analogous to the case in standard Text-to-Image models.
>
> ***“Can this method actually remove small parts of the image such as copyright logos? It was used in motivation but seems to not be tested?”***
>
> We aim to remove copyrighted content. The copyright law may vary between states, and may also contain the generation of artistic styles, fictional characters, or even letter fonts.
> Yet, generating small logos is hard to evaluate, as SD models typically would not produce them; neither would they significantly affect the CLIP score. In any case, our method can erase logos, similarly to any other object, but evaluation of small logos is difficult, and we do not claim it to be a use case special to our technique. We could not find a prompt that generates very small logos consistently (even without erasure). We are happy to evaluate any given prompt.
>
> We clarified this in the revised manuscript.
>
> ***“Does the approach work well also on other diffusion models than Stable Diffusion v1.4?”***
>
> We thank the reviewer for the suggestion. We provide additional results on Stable Diffusion 2.0 in Section C.3 of the Appendix. Our results suggest that TV can provide robustness against Concept Inversion for Stable Diffusion 2.0 on the Van Gogh concept.
>
> Thank you very much, once again, for your excellent comments. We added these additional experiments and explanations to the revised manuscript.
>
> We respectfully ask that if you feel more positive about our paper, to please consider updating your score. If not, please let us know what can be further improved; we are happy to continue the discussion any time until the end of the discussion period. Thank you!

---

> > ### Comment · Reviewer_kjLz · 2024-11-24
> > **Response**
> >
> > Thank you for the responses to my questions and concerns.
> >
> > Thank you for showing an example result with Stable Diffusion 2.0, but I am afraid showing only one example with SD2.0 does not make the response persuasive in showing the method works reliably. A quantitative analysis would be significantly more suitable.
> >
> > Not sure if we can say all issues with presentation have been resolved as an unusual font is still used. Currently I can also easily see other issues with presentation, in particular Table 5.
> > I recommend using a different colour than light green for the changes as this makes it a bit hard to read.
> >
> > Further, my concerns about the novelty of the method remain.

---

> ### Author Response · Authors · 2024-11-25
>
> Thank you for acknowledging our rebuttal and continuing the discussion.
>
> Following the reviewer's suggestion we include a quantitative evaluation of our results with SD2.0, following the evaluation setup similar to Table 2 in our submission (Acc. % with the given prompt / Acc. % with CI adversarial method):
>
>
> | Method         | SD 2.0 | TV  |
> |-----------------|----------------|---------------|
> | English springer | 95.1           | 11.2 / 0.1         |
> | Garbage truck            | 89.2          | 8.4 / 0.2         |
>
> We see that our method is erasing concepts effectively for SD2.0 as well. We include it along with additional qualitative results in the manuscript.
>
> The tables and fonts are now consistent (apart from table 2 which intentionally has smaller fonts). We also changed the color of the edits to purple per the reviewer’s request.
>
> Regarding novelty - We would like to emphasize that our analysis of the input complexity is novel, as well as using it to understand the importance of prompt independence. It motivates the technical solution we chose, which indeed relies on existing works that were previously used in other contexts.
> We believe that a well-motivated solution with a limited technical novelty may serve the community better than a solution optimized for novelty but not necessarily for effectiveness.
>
> Thank you once again for your suggestions for our work as well as your follow-up comment!

---

> > ### Comment · Reviewer_kjLz · 2024-11-25
> > **Thank you**
> >
> > Thanks for the new results and explanations. The paper looks better now and the results with SD2.0 seem sufficiently persuasive in showing the method works also on other models. I’ll keep an eye on the discussions with the other reviewers and may increase the rating later.

---

> ### Author Response · Authors · 2024-12-02
>
> Dear Reviewer,
>
> We appreciate your time and effort in reviewing our work and are pleased that you have a better view of it.
>
> We would like to kindly remind you that the discussion period is nearing its conclusion. One other reviewer has engaged with our response so far, and has increased their score to *8: accept, good paper*. The other two reviewers have not yet engaged with our rebuttal, but we believe that our responses address their concerns as well.
>
> We would greatly appreciate it if you could kindly reconsider your evaluation.
>
> Thank you very much for your time and consideration.

---

> > ### Comment · Reviewer_kjLz · 2024-12-02
> > **Increased score**
> >
> > Thanks, I've reviewed the discussions and decided to increase the rating of the paper.

---

### Official Review · Reviewer_5hfp · 2024-11-03

**Soundness:** 2
**Presentation:** 2
**Contribution:** 2
**Rating:** 5
**Confidence:** 4

**Summary:**

The paper presents an interpretability study focused on understanding the second-order effects of neurons in CLIP. The authors propose a novel "second-order lens" to analyze neuron contributions that flow through attention heads to the model output.

**Strengths:**

1. The technical contributions are sound  and interesting.
2. The paper is well written.
3. The paper included thorough evaluations.

**Weaknesses:**

1. Multiple concept erasure - How does the proposed method perform on multi-concept erasure? The baselines considered in this paper (UCE and ESD) evaluate their model on erasing multiple objects simultaneously. Therefore it is fair to compare this method for multi-concept erasure.
2. Missing baselines - Comparison to Selective Amnesia (SA) (a strong and very similar baseline in my opinion) is missing from the paper. I believe the proposed method lie under a similar umbrella as SA.
3. Underperforms baselines on NSFW concepts—The authors state that TV only reduces nudity in 52% of images compared to SD1.4, which is worse than the baselines (ESD, UCE, etc.) considered in the paper. This is a major drawback of the method in a real-world setting.

**Questions:**

See weaknesses

---

> ### Author Response · Authors · 2024-11-22
>
> Thank you for your detailed review. We appreciate that you found our work sound and interesting, and our evaluations thorough. We address each of your questions below:
>
> ***“...How does the proposed method perform on multi-concept erasure?...”***
>
> We include an evaluation of our methods when erasing multiple concepts at once. Fig 18 in the revised manuscript. We erase both Van Gogh and Thomas Kinkade by first taking the average of both task vectors and then subtract it from the original model. Moreover, for each concept we generate 50 images and compute the average CLIP score to provide quantitative results.
>
> | Concept         | SD 1.4 CLIP Score | TV CLIP Score | Concept Inversion CLIP Score |
> |-----------------|----------------|---------------|---------------|
> | Thomas Kinkade |  0.347        | 0.135         |	0.195	 |
> | Van Gogh            | 0.282          | 0.241        | 0.233		 |
>
> We would like to emphasize that our main improvement over UCE and ESD is in adversarial robustness. As demonstrated by our results, we maintain this prosperity in the multi-concept setting.
>
> ***“...Comparison to Selective Amnesia (SA) (a strong and very similar baseline in my opinion) is missing from the paper…”***
>
> We include additional results on performing Concept Inversion on a model with the Van Gogh concept erased using Selective Amnesia in Section C.3 of the Appendix. Our results suggest that Selective Amnesia does not exhibit robustness as well as our proposed method. This observation is done by [1]. Moreover, we would like to emphasize that while Selective Amnesia is a strong method, it is prompt-dependent. Therefore, as UCE, ESD, and similar methods, it indeed performs well against that prompt but is less robust to unexpected attacks.
>
> ***“Underperforms baselines on NSFW concepts … This is a major drawback of the method in a real-world setting”***
>
> While acknowledging this as a limitation of our method, we would like to emphasize that real-world settings differ from one another, and therefore might require different solutions. E.g., for maintaining copy-rights one may need robustness to unexpected prompts, which is a relative advantage of our method. We believe therefore that there is room for more than one method, until a single method would possibly incorporate all advantages.
>
> We add this discussion as well as the comparison asked by the reviewer to the revised version of the manuscript. Thank you very much, once again, for your excellent comments. We respectfully ask that if you feel more positive about our paper, to please consider updating your score. If not, please let us know what can be further improved; we are happy to continue the discussion any time until the end of the discussion period. Thank you!
>
> **References**
>
> [1] Minh Pham, Kelly O. Marshall, Niv Cohen, Govind Mittal, Chinmay Hegde. “Circumventing Concept Erasure Methods For Text-To-Image Generative Models”, ICLR 2024

---

> ### Author Response · Authors · 2024-11-27
>
> Dear Reviewer,
>
> We sincerely appreciate your valuable feedback and the time you have dedicated to reviewing our submission. Your insights have been instrumental in shaping the final version of our submission.
>
> We would like to kindly remind you that the discussion period is set to conclude on December 2nd. If there are any additional questions, concerns, or clarifications, we would be delighted to continue the discussion.
>
> Thank you once again for your attention. We look forward to hearing from you!

---

### Author Response · Authors · 2024-11-22

We thank all the reviewers for their valuable feedback. Our work highlights the dependence on specific prompts as a source of fragility in concept erasure methods, and studies task vectors as a prompt-independent solution. We appreciate that the reviewers find our contributions well-motivated (*kjLz*, *EQBm*, *yv46*), the results sound and interesting (*5hfp*, *yv46*), and the paper clear and well-written (*yv46*, *EQBm*, *kjLz*, *5hfp*).

Following your suggestions, we highlight further improvements:

**(A) We add additional quantitative results for UnlearnDiffAtk and P4D**

We evaluated the robustness of the erased model against these two attacks over 10 iterations. Please find the quantitative results below:

| Method         | ESD CLIP Score | TV CLIP Score |
|-----------------|----------------|---------------|
| UnlearnDiffAtk | 0.265          | 0.211         |
| P4D            | 0.258          | 0.204         |

We also added this result to the revised manuscript.

**(B) We extend the evaluation of our method's applicability to  SD 2.0**

We validated that the Task Vector technique indeed supplies robust erasure of concepts on the model as well.

Results appear in the revised manuscript in App C.3 Fig.16.

**(C)  We incorporate additional baselines as Selective Amnesia (SA)**

We incorporated Selective Amnesia as an additional comparison. Similarly to [1] we find that Selective Amnesia is not robust against attacks.

Results appear in the revised manuscript in  App C.3 Fig.17.

Additional answers to individual reviewer concerns are detailed below. We would be very happy to keep the discussion going, addressing any points that remain unclear, or any new suggestions. Thanks again for your suggestions!

**References:**

[1] Minh Pham, Kelly O. Marshall, Niv Cohen, Govind Mittal, Chinmay Hegde. “Circumventing Concept Erasure Methods For Text-To-Image Generative Models.” ICLR 2024

---

### Public Comment · ~Finn_Carter1 · 2024-11-22
**Lack of recent related works**

It seems that many recent related works like [1,2,3,4] are ignored in this work.

[1] One-dimensional Adapter to Rule Them All: Concepts, Diffusion Models and Erasing Applications

[2] MACE: Mass Concept Erasure in Diffusion Models

[3] Receler: Reliable Concept Erasing of Text-to-Image Diffusion Models via Lightweight Erasers

[4] Separable Multi-Concept Erasure from Diffusion Models

---

> ### Author Response · Authors · 2024-11-22
>
> Dear Finn Carter,
>
> Thank you for bringing these works to our attention. While they study similar settings, they are not as focused on understanding adversarial robustness as our work. The Receler method [3] does study robustness, but it does not explore robustness against soft prompt attacks which is a main topic in our paper. Nevertheless, we agree that these works are relevant, and we have revised our manuscript to incorporate them as related works.

---

### Author Response · Authors · 2024-12-03
**Summary of Paper Revisions**

We thank all reviewers for their time and effort invested in reviewing our paper, and their useful feedback. Below, we provide a summary of the main updates we made:

- Added an evaluation for an additional diffusion model (SD 2.0)
- Validated the effectiveness of our technique in erasing multiple concepts
- Added results evaluating additional attacks and baseline methods

The final version will incorporate all suggestions not yet included in the revised version.

Once again, we sincerely thank the area chair and all reviewers!

---

### Meta-Review · Area_Chair_fKcc · 2024-12-19

**Metareview:**

This paper introduces a novel approach to concept erasure in text-to-image diffusion models. It focuses on a prompt-agnostic framework and accomplishes this by leveraging task vectors and a variant of textual inversion. The methodology is not novel, but the concept is interesting and can inspire future work in the field.

However, as the authors also stated, "We would like to highlight identifying the prompt dependence as the source of vulnerability in previous concept erasure methods. This is a main contribution of our work," admitting the methodology is not their main contribution, sufficient experimental evaluation should be observed as evidence for their argument. The experimental results fall short, demonstrating limited exploration of different base models. Although the added experiments with SD2.0 somewhat evidenced the argument on other models rather than SD1.4 alone, the evaluation is not thorough enough to make the paper stand. Furthermore, compared to existing baselines, the proposed method underperforms in critical real-world use cases, such as NSFW content erasure. This weakens its practical utility, particularly for sensitive applications where reliability is significant.

**Additional Comments On Reviewer Discussion:**

Two reviewers were happy with the authors' rebuttal, while one did not join the discussion. Although Reviewer 5hfp did not respond to the rebuttal, AC agreed that the failure in the NSFW evaluation, along with other concerns, dragged down the paper's solidity.

---

### Decision · Program_Chairs · 2025-01-22

Reject